# Micro-EDM Drilling/Milling as a Potential Technique for Fabrication of Bespoke Artificial Defects on Bearing Raceways

**DOI:** 10.3390/mi13030483

**Published:** 2022-03-20

**Authors:** Long Ye, Krishna Kumar Saxena, Jun Qian, Dominiek Reynaerts

**Affiliations:** 1Micro- & Precision Engineering Group, Department of Mechanical Engineering, KU Leuven, 3001 Leuven, Belgium; long.ye@kuleuven.be (L.Y.); krishna.saxena@kuleuven.be (K.K.S.); jun.qian@kuleuven.be (J.Q.); 2Corelab@KUL-MaPS, Flanders Make, 3001 Leuven, Belgium

**Keywords:** artificial bearing defects, micro-EDM, bearings, tribology

## Abstract

The fabrication of bespoke artificial defects on bearing raceways helps in mimicking incipient faults during real application or for directly validating the diagnostic technology depending on their shapes and sizes. This is particularly useful when run-to-failure experiments are time-consuming and even difficult in some cases. However, there has been limited systematic research on the design and fabrication of artificial defects on bearing raceways, particularly for the purpose of accelerated testing. In this work, micro-EDM is put forward as a potential technique for the fabrication of artificial defects using drilling/milling mode. A methodology is developed, not only to achieve the full control of the dimension and distribution of defects on a bearing element, but also to qualitatively and quantitatively perform the efficient characterization of the defect surface. A linear regression model with the inclusion of two-way interactions based on an analysis of variance (ANOVA) is presented to optimally select the process parameters. The verification experiments show that this mathematical model obtains a good fit for approximately 80% of the observed data. Through a combination of optical microscopy and confocal microscopy, the morphology and topography of the artificial defects was measured and compared. To conclude, micro-EDM evidences its great potential in terms of machining efficiency, e.g., with an MRR of 0.060 ${\mathrm{mm}}^{3}/\min$, TWR of 0.032 ${\mathrm{mm}}^{3}/\min$ and dimensional controllability, e.g., the standard deviation of pitting diameter and depth being 0.5 µm and 0.8 µm, respectively, to achieve a desirable feature shape for bearing defects.

## 1. Introduction

Rolling/ball element bearings are indispensable components which are widely used for rotating machines. On the other hand, they are susceptible to failure due to deteriorated operating conditions and reduced fatigue strength, leading to unexpected machine downtime and eventually, environmental pollution and economic loss. Therefore, the techniques for bearing condition monitoring are employed based on either vibration signals [1] or acoustic signals (AE) [2] which are collected through run-to-failure or degradation tests. However, these tests are time-consuming and thus are not applicable to the verification of a certain detection technology, for instance, a machine learning-based method, which requires a huge training set. For this reason, artificial defects are often induced in the form of indentation, line spalling or square wear to shorten the incipient period of growing bearing faults.

Although artificial defects are extensively used in bearing fault diagnostics and prognostics, their fabrication technique as well as shape and dimension have not been standardized. To investigate the AE technique for identifying bearing defects, Al-Ghamd et al. introduced point defects, line defects and big rough defects in dimensions of 0.85 × 0.85 mm, 5.6 × 1.2 mm and 17.5 × 9.0 mm, respectively, on an outer raceway for simulating varying severity [3]. The line defect was further tested with seven sizes to authenticate the observations related to AE. Similarly, Hemmati [4] studied two fault geometries, i.e., a radial notch and a narrow line, on different bearing locations to mimic real defects. In their work, the width of a line defect was 1 mm or 2 mm while the diameter of a radial notch was 1 mm or 2 mm. To demonstrate the application of a fiber-optic sensor in estimating fault size, Alian et al. created artificial line-spalling defects in ten different widths ranging from 0.39 mm to 4 mm on both outer and inner rings [5]. The fault location was distributed along 12 angular orientations to differentiate the load. 

Various alternative processing techniques for producing bearing defects have been reported in prior state of the art studies. Al-Ghamd et al. [3] and Hemmati [4] engraved the artificial defects while Alian et al. [5] produced the defects with EDM. Additionally, laser machining has also been reported to manufacture bearing faults [6]. In view of the difficulty in maintaining the geometric consistency when applying engraving and laser machining, EDM seems to be more promising in practically and economically producing accurate artificial bearing defects. Its non-contact machining traits enable machining on bearing races, independently of hardness. Chen et al. [7] used die-sinking EDM to seed consistent and measurable line spalling with a width of 0.794 mm, 1.135 mm and 1.530 mm onto the bearing outer race. The line defect was maximized in length and depth so as to support the estimation model. Similarly, the seeded line defects produced by Alian et al. [5] were also varied in width, ranging from 0.39 mm to 4 mm, either onto the bearing outer ring or bearing inner ring. This fact proves that EDM has a great capability to achieve the variable defect size and suitability in the defect shape control. As a variant of EDM, micro-EDM has the capability to manufacture miniaturized components with features in the submillimeter scale and with a machining accuracy of 2–3 microns [8]. This paper reports on the application of micro-EDM drilling and milling to artificially introduce pitting, line-spalling and notch bearing defects. Since the machining efficiency is subject to process stability, a design of experiments (DoE) was conducted to select the optimal process parameters. Confocal measurements of the profile and topography of the machined defects are also presented.

## 2. Materials and Methods

In this research, the fabrication of artificial defects was carried out using micro-EDM on an SKF^®^ spherical roller thrust bearing and an SNFA^®^ ball bearing made of SKF Xbite heat-treated steel and SAE 52,100 chrome steel, respectively. The experimental campaign was conducted on a standard Sarix^®^ SX-100-HPM machine. In the milling set-up, as shown in Figure 1a, a wire electrical discharge grinding (WEDG) device was attached to fabricate an electrode down to 10 µm in diameter. The wire between two plates was used for roughing, after which the wire transferred around the blocks was used for finishing. As can be seen from the left-side window of Figure 1a, a single-station vise along the *Y* axis was fixed by two step set-up clamps, which were bolted into the tooling plate. The ball bearing element fixed on the *Y* axis can be tuned to any angle relative to the *X* axis to avoid collision with the milling spindle. The yellow tube provided external flushing during the process. HEDMA oil was used as a dielectric in this experiment. In the top of the picture, a spindle, with a rotation speed of up to 600 RPM, was attached to the frame through a System 3R chuck. The tool length was maintained constant through automatic electrode feeding where the wear of tool length was measured by electrical touch on the workpiece surface. In order to have minimal tool wear, a tungsten carbide rod was used as tool electrode. In the drilling set-up, as shown in Figure 1b, an interchangeable drilling collet was used instead. Since the tool working length is quite long in a drilling regime, a ceramic tool guide was employed to reduce the tool run-out. 

The drilling and milling strategies were then determined for the generation of artificial pitting and spalling, respectively. As shown in Figure 2a, the artificial pitting can be directly drilled onto the bearing inner or outer ring surface. Considering that the diameter of pitting is varied depending on specific applications, a planetary drilling strategy with the inclusion of a lateral gap (overcut at approximately 5 µm in this pitting case) was employed. To further ensure the depth accuracy, a longitudinal tool wear compensation was implemented on the basis of material investigations from the preliminary experiment (Section 3). As for generating artificial spalling, as presented in Figure 2b, a uniform compensation strategy [9], i.e., reciprocating the tool trajectory for each two consecutive milling layers, was employed to counteract the dimensional inaccuracy in the *x* axis caused by the tool wear. The tool wear left merely in the *z* axis was then compensated in a way analogous to the strategies used for drilling. This way, the spalling dimensions were precisely controlled.

The generated defects were characterized by a 3D optical profilometer, namely Sensofar^®^ S Neox. After characterization, the removal volume of each defect was able to be computed based on a delimitation method, as illustrated in Figure 3. Concretely, in the first step, an outline was drawn covering the pitting area. In the second step, the outline was segmented into numerous small vessels, each upper topography of which was defined as the nominal surface while each lower topography of which was set as the bottom outline, as presented in Figure 3c. Finally, the removal volume of each vessel was computed based on the upper and lower topography and their summation constitutes the overall material removal of one defect. To calculate the tool wear rate (TWR), the longitudinal wear was measured on-machine by conducting an electrical touch between the working electrode and the referenced pin before and after each process of defect generation.

## 3. Preliminary Experiments

Micro-EDM is by nature a thermoelectric process in which materials are eroded by a series of sparks. The control of the sparking phenomena involves a lot of parameters, e.g., pulse frequency, pulse on/off time, voltage and current. It is thus essential to explore an optimized combination of process parameters with respect to specific bearing materials. The EDM performance indices that were used to evaluate the removal process were taken to be the material removal rate (MRR) and TWR, which are calculated as below:(1)$$\begin{array}{c} \mathrm{MRR}\left(\frac{{\mathrm{mm}}^{3}}{s}\right)=\sum_{i=1}^{n}v_{i}/t\\ \mathrm{TWR}\left(\frac{{\mathrm{mm}}^{3}}{s}\right)=\pi r^{2}\times \Delta l/t \end{array}$$
where r denotes the radius of electrode; Δl denotes the longitudinal tool wear; t is the machining time; $v_{i}$ is the removal volume of each small vessel; and n is the number of the vessels wrapping up the cropped outline. Since a conventional numerical model [10] is not very suited to accurately predict the MRR and TWR, a statistical model based on an analysis of variance (ANOVA) was employed in this paper. In accordance with the control parameters of our pulse generator (SARIX^®^ PULSAR), the pulse frequency, current, regulation gain and servo voltage were selected as the main factors and their levels are shown in Table 1. It is worth mentioning that the regulation gain is representative of the servo control reaction. The bigger the regulation gain, the higher feed-rate the system may achieve but the greater instability it can be subject to. In order to perform the experiments within reasonable time for screening processing parameters, a two-level full factorial design was conducted by means of two repeated runs for each axial point and five runs for central point. The run order was then randomized in order to balance uncontrollable effects.

ANOVA results for two-factor interaction (2FI) models when responses are taken as MRR and TWR are listed in Table 2. The *p*-values of the main effects and 2FI terms are both close to 0, although the exact values of the main effects are slightly smaller than those of 2FI. This suggests that both the main effects and 2FI terms are significant to be incorporated into our fitting model. In addition, the model adequacy can be inferred from residual analysis. The F-statistic, as a ratio of the lack of fit mean square and pure error mean square, results in a large *p*-value, i.e., 0.15 for MRR or 0.23 for TWR. It implies that the lack of fit for our 2FI model is insignificant, i.e., the 2FI model can sufficiently explain the variability because of the negligible effects that are pooled into the residual error. Furthermore, a main effects plot, as shown in Figure 4, is used to quantitively estimate the statistical significance of each factor variable. In general, the slope of an effect line is proportional to the significance of that factor. It is thus concluded that the pulse frequency is the most significant factor in influencing both the MRR and the TWR, while the current is the least important factor. This is because in this retrofitted pulse generator, the current may be inherently varied with respect to a predefined energy level. 

The final regression model only incorporates the main effects and 2FI terms whose *p*-values are lower than the significance level, i.e., 0.05 in this paper. On this basis, the $R^{2}$ statistics [11] calculated are 81.90% and 82.76%, respectively, for the MRR and the TWR, which indicates that approximately 80% of the observed variation could be explained by main effects and two-way interactions. The coded prediction model for the MRR and the TWR are as follows:(2)$$\hat{\mathrm{MRR}}\left({\mathrm{mm}}^{3}/\min\right)=0.0348-0.0126\;A-0.0079\;C\;+0.0065\;D\;-0.0059\;\mathrm{AB}\;+0.0064\;\mathrm{AC}\;-0.0074\;\mathrm{CD}\;\hat{\mathrm{TWR}}\left({\mathrm{mm}}^{3}/\min\right)=-0.0229-0.0122\;A\;-0.0055\;C\;+0.0031\;D\;+0.0027\;\mathrm{AC}\;+0.001\;\mathrm{BC}\;-0.0064\;\mathrm{CD}$$

On the basis of Equation (2), an optimization of maximizing the MRR as well as minimizing the TWR is achieved by virtue of a multiplicative desirability function [12], which is a representative of the distance between the searched responses and the preset boundary. As such, the MRR of 0.060 ${\mathrm{mm}}^{3}/\min$ and the TWR of 0.032 ${\mathrm{mm}}^{3}/\min$ can be achieved in the case of a pulse frequency of 60 kHz, a current of 140, a regulation gain of 60 and a servo voltage of 70 volts. 

## 4. Results and Discussion

The validation test for the optimal parameters was conducted on the generation of repeated pittings, whose nominal diameter and depth are expected to be 400 µm and 80 µm, respectively. The characterized results are shown in Figure 5. It can be observed that a consistent pitting topography in terms of diameter and depth can be achieved by micro-EDM. This implies that, the optimally selected parameters contribute to a stable removal process for bearing materials. The diameter and depth of each pitting are estimated and shown in Table 3. It can be seen that the standard deviation of the pitting diameter and depth are 0.46 µm and 0.85 µm. A higher variance in the pitting depth may be attributed to the difficulty of estimating and controlling the longitudinal tool wear and the frontal gap in the same micro-EDM process. Similarly, the calculated MRR and TWR with respect to each validation test are listed in Table 3. The average values are in excellent accordance with those estimated by the above 2FI model.

The bearing defect usually grows from a surface imperfection in the incipient phase to a damaged area in the failed phase. A typical example is presented in Figure 6 in which the three stages of a defect on the inner ring are varied during an accelerated test. It is visually obvious that the defect expands not only along the raceway but also inwards radially. This might be the result of uneven stress on the defected part when running. The initial defect in this test was manually induced by indentation using a Vickers hardness tester. However, this tester is restricted in precisely maintaining the artificial defect feature and distribution compared to the manufacturing capabilities of a micro-EDM machine.

To precisely compare the topography of the induced pitting and indentation, confocal measurements shown in Figure 7 were performed. Since the diamond indenter used in the Vickers hardness tester is conical, the profile of indentation resembles a round V shape. Although the depth of the pitting is the same as that of the indentation, its profile resembles a U shape because of the inevitable corner wear showing up on the cylindrical electrode when machining. Furthermore, by changing the process parameters introduced in Section 3, the bottom corner angle and outline can be tuned with respect to the topography of a running defect. In addition, it can be seen that there is an obvious blur around the indent edge left by the indenting process, whereas the pitting edge seems smooth and consistent after optimization by the micro-EDM process. This thus avoids the creation of topological effects that cannot be quantitively evaluated upon acceleration test results.

The machined defects on the bearing interacting surfaces of each element are shown in Figure 8. The pittings on the thrust ball element shown in Figure 8a are uniformly distributed along the centerline. The diameter and depth of each pitting are approximately 550 µm and 80 µm with a standard deviation of 0.5 µm and 0.8 µm, respectively. The pittings shown in Figure 8b were produced on the bearing raceway with the same size as that of Figure 8a. These machined faults are comparatively small in order to mimic an incipient defect thereby enabling the observation of it evolving under different operating conditions. However, in contrast to the singular defects, as can be seen in Figure 7, the distributed defects correspond to the case in which the incipient defects on a localized surface position are spread over a wide surface area or over the complete surface, thereby giving rise to continuous vibration upon a bearing. As a comparison, the spalling with a width of 500 µm and length of 5 mm, as shown in Figure 8c, was radially machined onto the inner raceway. This artificial feature is relatively large in order to mimic a running fault in the grown stage, thereby accelerating the validation of newly developed detection technologies. The same spalling feature was also copied to the outer ring, as seen in Figure 8d. However, due to the difficulty of the direct characterization of the spalling on the outer ring, we employed a defect-replicating technique [13] (replica compound of silicone polymers Microset^®^ 101 RF, resolution of 0.1 µm). Concretely, the silicon polymer was first injected to cover the spalling feature. Then, a backing-slide was used to slightly press the injected polymer facilitating the uniform distribution of the injected polymer. Once the polymer was cured to form a reverse replica of the spalling, it can be carefully pulled out. Finally, the measurement of the replica gives a precise characterization of the spalling geometry. It is obviously observed that by adopting the contour-parallel milling strategy (presented in Figure 2c,d), the produced spalling has a uniform depth along its direction of growth, and was thus able to offer a consistent defect frequency for the use of either phenomenological or data-driven modeling. Being able to mimic the main features of surface fatigue that leads to bearing failure under proper lubrication and operating conditions, this bearing defect was especially designed for the test rig which was later used for an estimation of the bearing’s remaining useful life.

As shown in Figure 9, the randomly generated pittings on a specific area of the bearing outer surface were also generated for the purpose of mimicking. This distribution was due to the fact that some pittings are much more likely to appear in a group when the lubrication is highly contaminated by exotic particles. By using a post-program before drilling, a set of pittings with a pre-defined hole area ratio, e.g., 15% in this paper, are irregularly generated. The generated pittings are then re-located in a user-defined reference and their coordinate information is sent to an operating program in the EDM machine for automatic drilling. In Figure 9, all the pittings are distributed with the same diameter and depth, i.e., 500 µm and 70 µm, respectively. However, the pittings can also be produced with varying topologies when using WEDG for fabricating the tool electrodes. Since there is a restriction applied on the gap between holes, all pittings were observed separately. By taking into account the randomness of the distributed pittings, we demonstrated the capabilities of EDM in generating the artificial bearing defects to be more comparable to natural defects. 

## 5. Conclusions

The potential of using micro-EDM drilling/milling to fabricate artificial bearing defects (spalling, pitting) on bearing elements (outer race, inner race and ball element) has been addressed in this work. A dedicated drill/mill strategy with tool-wear control was employed for machining on each bearing element (inner race, outer race and ball). By using a linear regression model based on an ANOVA analysis, the process parameters were optimally selected to achieve a high machining performance and ideal feature topography. The validation test confirmed the suitability of the optimal parameters in terms of ensuring the defect dimensions, e.g., the standard deviation of diameter and depth being 0.46 µm and 0.85 µm, respectively, as well as maintaining a high accordance of MRR and TWR with their estimations from the 2FI model. The pitting and spalling seeded onto either bearing’s inner race or outer race further demonstrate the micro-EDM capability in generating customized artificial defects on the geometrically complex surface with a tight tolerance. Future work was planned to characterize the sub-surface cracks introduced by the artificial bearing defects and investigate what effects they will have on a running test.

## Figures and Tables

**Figure 1 micromachines-13-00483-f001:**
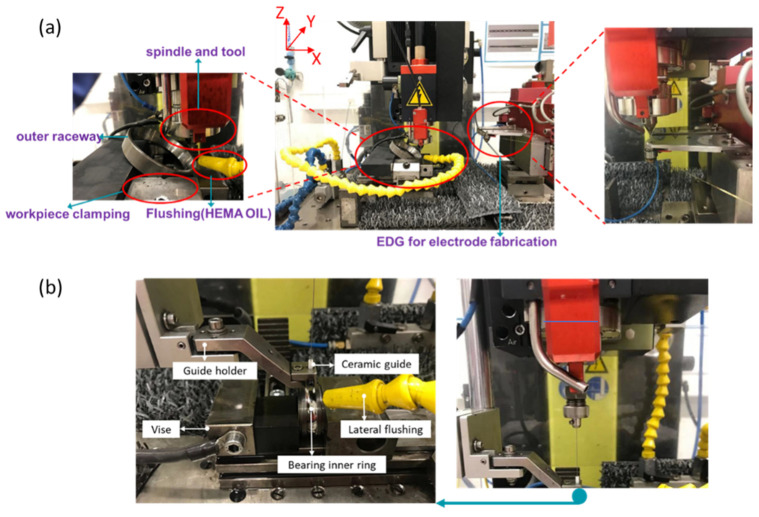
Micro-EDM (**a**) milling and (**b**) drilling set-ups for artificial defects machining.

**Figure 2 micromachines-13-00483-f002:**
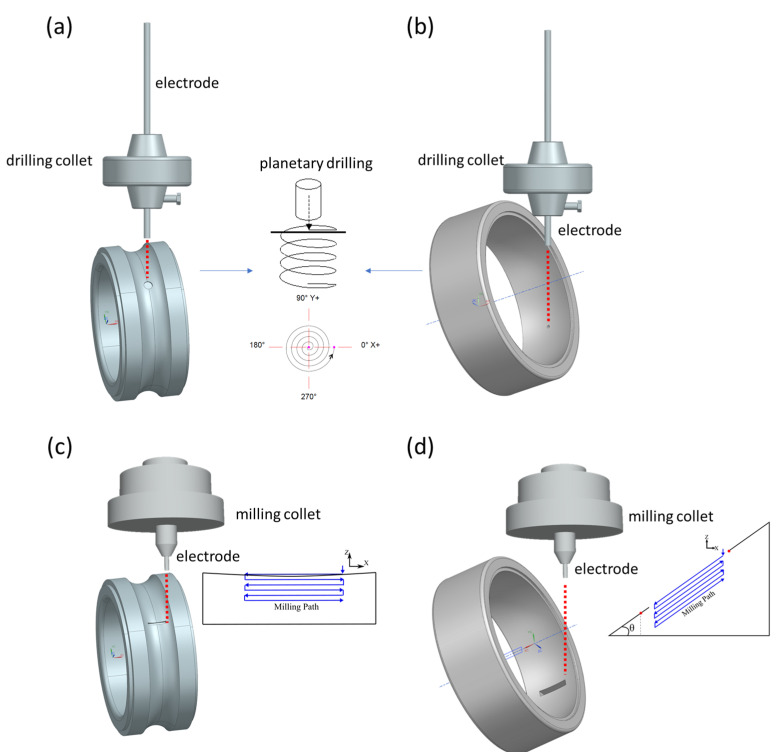
Micro-EDM drilling and milling strategies for generating artificial defects. Drilling of artificial pitting onto the (**a**) bearing inner ring and (**b**) bearing outer ring; milling of artificial spalling onto (**c**) bearing inner ring and (**d**) bearing outer ring. (Sketch not to scale).

**Figure 3 micromachines-13-00483-f003:**
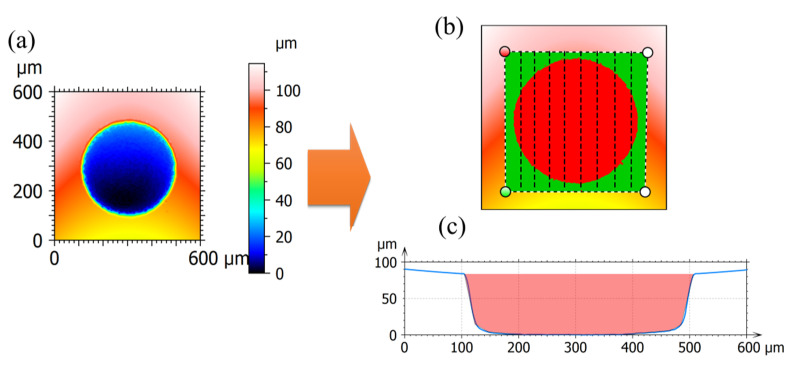
The calculation of the material removal volume based on the optical measurement. (**a**) the pitting topography by measurement; (**b**) A mask consisting of numerous segments to cover the pitting area; (**c**) the cross-sectional profile where the pitting volume is filled.

**Figure 4 micromachines-13-00483-f004:**
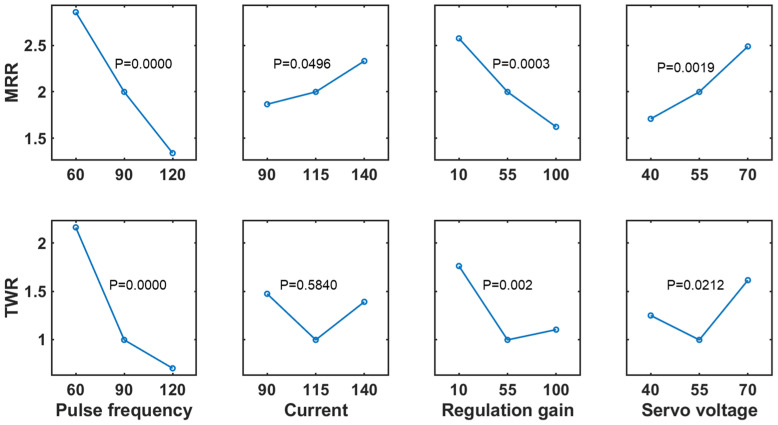
The main effect estimate of input factors, i.e., the pulse frequency, current, regulation gain and servo voltage with respect to the MRR and TWR responses. The *p*-value from the ANOVA analysis is given on top of each plot to illustrate the significance of each parameter.

**Figure 5 micromachines-13-00483-f005:**
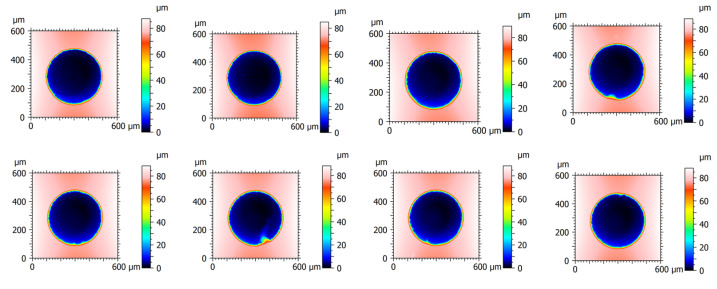
Validation test with repeated pittings on the bearing inner ring.

**Figure 6 micromachines-13-00483-f006:**
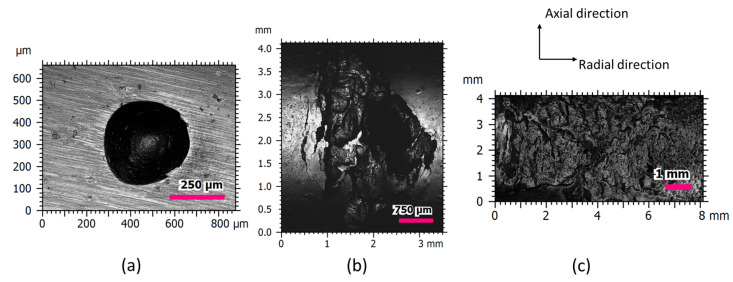
The characterization of running bearing defects. (**b**,**c**) are defects grown from (**a**) indentation with increasing running cycles under the same condition. Measurement were performed by Sensofar S Neox, with an objective of 20×, restoration applied and no filtering applied.

**Figure 7 micromachines-13-00483-f007:**
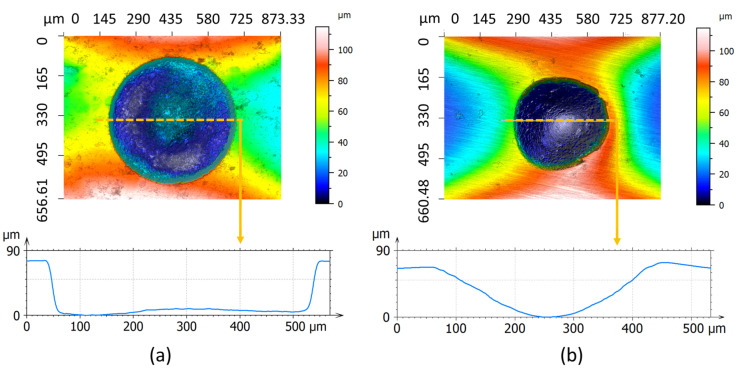
2D/3D topography and the profile of artificial defects (**a**) pitting produced by micro-EDM (**b**) indentation produced by Vickers hardness machine. Measurement by Sensofar S Neox, with an objective of 20×, form removal and restoration applied and no filtration applied.

**Figure 8 micromachines-13-00483-f008:**
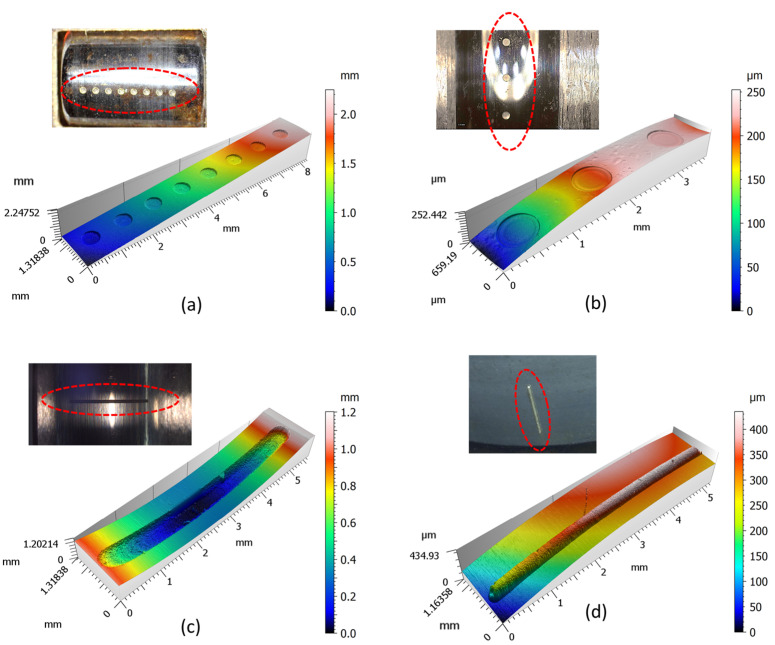
Uniformly distributed pittings (**a**) on a thrust ball element and (**b**) on a bearing inner raceway. Singular spalling (**c**) on a bearing inner raceway and (**d**) on a bearing outering (replica).

**Figure 9 micromachines-13-00483-f009:**
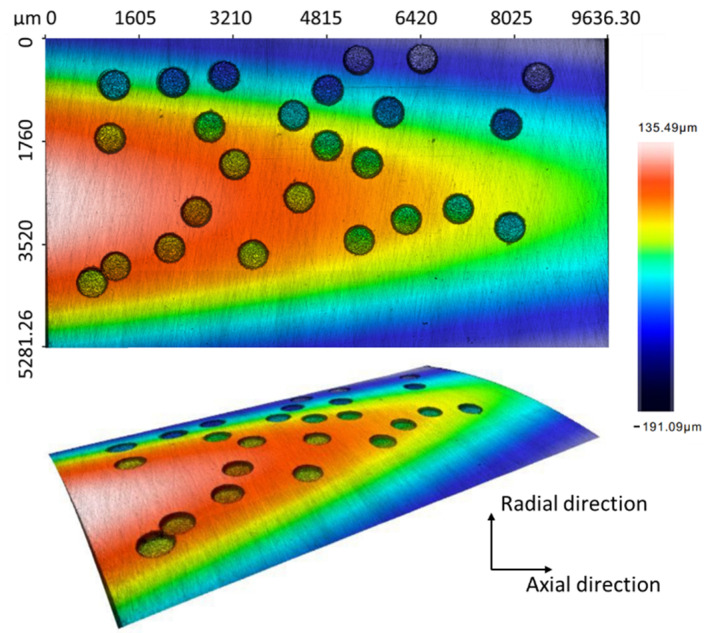
Random pittings generated by micro-EDM on the bearing’s outer ring. Measurement was performed by Sensofar S Neox, with an objective of 20×, restoration applied and no filtration applied.

**Table 1 micromachines-13-00483-t001:** Input factors with their levels for full factorial design.

Factors	Abbreviation/Notation	Unit	Levels	
			Low (−)	High (+)
Pulse frequency	Freq/A	kHz	60	120
Current	Cur/B	Index	90	140
Regulation gain	K/C	Index	10	100
Servo voltage	Ve/D	Volts	40	70

**Table 2 micromachines-13-00483-t002:** The two-way ANOVA results.

Effects	Degree of Freedom	Sum of Squares		F-Value		*p*-Value		R^2^	
		MRR	TWR	MRR	TWR	MRR	TWR	MRR	TWR
Main effects	4	0.0089	0.0061	18.5417	30.5000	0.0000	0.0000	81.90%	82.76%
Two-way interactions	6	0.0043	0.0017	5.9722	5.6667	0.0005	0.0007		
Residual	26	0.0029	0.0013						
Lack of fit	6	0.0010	0.0004	1.7895	1.4815	0.1500	0.2300		
Pure error	20	0.0019	0.0009						

**Table 3 micromachines-13-00483-t003:** Validation test results.

Test Number	Diameter/µm	Depth/µm	MRR/(mm^3^/min)	TWR/(mm^3^/min)
1	400.12	78.64	0.058	0.033
2	401.06	80.66	0.058	0.033
3	399.75	80.19	0.063	0.029
4	399.91	80.88	0.060	0.029
5	399.73	80.55	0.063	0.030
6	399.61	79.32	0.057	0.032
7	400.09	78.87	0.060	0.031
8	400.25	79.95	0.063	0.031
Mean	400.07	79.88	0.060	0.031
Standard deviation	0.46	0.85	0.002	0.002

## Data Availability

The data on accelerated texting of bearings are currently the confidential data of the FlandersMake DGTwinPrediction project consortium.

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
