# Peer review of "Micro-EDM Drilling/Milling as a Potential Technique for Fabrication of Bespoke Artificial Defects on Bearing Raceways"

_micromachines, 2022, doi:10.3390/mi13030483_

Round 1

Reviewer 1 Report

  1. Label the axis in the Figure 1(a) as this would greatly help to understand the description in the manuscript.
  2. The method section does not clearly explain the measurement of tool wear rate and material removal rate. In this section, the author should describe how the TWR and MRR were collected and any instruments used.
  3. Amend the format of the captions for Fig.3-Fig.7
  4. Could not get the MRR of 0.058 mm3 /min and the TWR of 0.036 mm3 /min in the case of pulse frequency of 60 kHz, current of 140, regulation gain of 60 and servo volt of 70 volts from the equation (2). Kindly check.
  5. It would be good to show the R-sq obtained from ANOVA in Table 2.
  6. Advise to conduct a confirmation test for the optimal TWR and MRR obtained from the propose model.
  7. In line 224, typo error for purpose
  8. No comparison with the real defects from the previous studies. I would like to see some discussion of the findings of the papers in relation to previous findings.

Author Response

Please find all my responses in the attachment. Thanks a lot!

Reviewer 2 Report

The current paper reports the use of micro-EDM technique for introducing artificial defects for accelerated testing. The current manuscript lacks proper background of the current study, the experimental section needs modification. The result/discussion section needs to be revised with proper conclusions in view of additional data. The dimensions of the pits with respected to input parameters must be added in a Table. Based on my assessment, I suggest major revision of this paper.

The other comments are as follows:

  1. Line 47: Ref is missing.
  2. In the introduction section, the authors have mentioned the role of micro-defects by various techniques. What is missing is that, a more comprehensive background and associated outcomes from EDM, in general, if available in literature, on that.
  3. Please move the Table 1 in section 2.
  4. From my point of view, three variable for a given parameter (for example three different pulse frequency for TWR/MRR etc.) is not enough for a confident outcome of the results. If possible, then I suggest to carry out additional experiments with additional parameters.
  5. 6: As the surface is curved, the colour code regarding depth will be wrong. I think there should be a function in the software to adjust that.
  6. Also, what I am missing, the overall statistical analysis of the outcomes with error and significant of confidence.
  7. 7: Numerical values of the depth and diameters of the pits together with statistically significant and in relation to the experimental parameters must be included.
  8. Conclusion section needs to be re-written. In the current form, its just the description, what has been done in this work. I don’t find any conclusion based on the experimental outcomes!
  9. In the title the authors stated “……………for accelerated testing” . This gives the impression that, the authors are also conducting and exporting the results of accelerated testing; however, which was not the cases! Thus, the authors either needs to revisit the title or include the results of accelerated tests.

Author Response

(The authors gave the same response as above.)

Round 2

Reviewer 2 Report

  1. The title is not still corrected!